# Bergamot Polyphenol Extract Reduces Hepatocyte Neutral Fat by Increasing Beta-Oxidation

**DOI:** 10.3390/nu14163434

**Published:** 2022-08-20

**Authors:** Angela Mirarchi, Rosario Mare, Vincenzo Musolino, Saverio Nucera, Vincenzo Mollace, Arturo Pujia, Tiziana Montalcini, Stefano Romeo, Samantha Maurotti

**Affiliations:** 1Department of Medical and Surgical Sciences, University “Magna Græcia” of Catanzaro, 88100 Catanzaro, Italy; 2Department of Clinical and Experimental Medicine, University “Magna Græcia” of Catanzaro, 88100 Catanzaro, Italy; 3IRC-FSH-Pharmaceutical Biology-Department of Health Sciences, University “Magna Græcia” of Catanzaro, 88100 Catanzaro, Italy; 4IRC-FSH Department of Health Sciences, University “Magna Græcia” of Catanzaro, 88100 Catanzaro, Italy; 5Research Center for the Prevention and Treatment of Metabolic Diseases, University “Magna Græcia”, 88100 Catanzaro, Italy; 6Department of Molecular and Clinical Medicine, Institute of Medicine, Sahlgrenska Academy, Wallenberg Laboratory, University of Gothenburg, 40530 Gothenburg, Sweden

**Keywords:** citrus bergamia, human liver organoid, NAFLD, nutraceutical, spheroids

## Abstract

Background: Bergamot polyphenolic fraction (PF) extract exerts a beneficial against liver steatosis. However, the fundamental processes underlying this beneficial effect of bergamot PF remain elusive. In this work, we examined the effect of bergamot PF extract on 2D and 3D hepatocyte cultures. Material and Methods: We evaluated the effect of bergamot PF in 2D and 3D cultures from rat, human hepatoma cells, and human primary hepatocytes. Results: In 2D cell culture, we demonstrated that incubation with bergamot PF decreases intracellular lipid content and is associated with an increase in expression levels of ß-oxidation genes (*Acox1, Pparα*, and *Ucp2*) and lipophagy (*Atg7*). Moreover, we confirm this effect on 3D spheroids and organoids. Conclusion: Incubation with bergamot PF reduces intracellular lipid neutral fat potentially by increasing intracellular pathways related to beta-oxidation.

## 1. Introduction

Non-alcoholic or metabolic associated fatty liver disease (NAFLD) is the most common cause of chronic liver disease, and it is set to become the leading cause of liver transplants [1,2]. Specifically, NAFLD is a spectrum of conditions ranging from simple fat accumulation to inflammation, fibrosis and, ultimately, cirrhosis and hepatocellular carcinoma [1]. Metabolic disorders including insulin resistance are key risk factors promoting hepatic lipid accumulation and, therefore, NAFLD is also designated as metabolic dysfunction-associated fatty liver disease [2]. Furthermore, NAFLD has a strong genetic component with a handful of common genetic variants conferring an increased risk of onset and progression of NAFLD [3,4]. Mendelian randomization studies using these genetic variants have demonstrated that the excess in liver fat content per se causes liver cirrhosis and hepatocellular carcinoma [5,6]. Importantly, after more than a decade of efforts, there is no effective pharmacological treatment specifically approved against NAFLD.

*Citrus bergamia risso et poiteau* (Bergamot), the fruit from a tree growing in Calabria, a southwest region of Italy, has a high content of flavonoids and glycosides [7,8]. Moreover, bergamot contains melitidin and brutieridin, two molecules with statin-like effects [9,10,11], as well neoeriocitrin, narigin, and neohesperidin, which are polyphenolic compounds. Studies in humans [12] and rodents show that administration of a bergamot polyphenolic fraction (PF) reduces liver fat content. Administration of bergamot PF in rats fed a cafeteria diet resulted in lower hepatic triglycerides content, possibly due to an increase in autophagy [13]. However, the fundamental processes underlying this beneficial effect of bergamot PF extract remain obscure. In this work we evaluated the effect of bergamot PF in 2D and 3D cultures from rat, human hepatoma cells, and human primary hepatocytes on intracellular lipid content.

## 2. Materials and Methods

### 2.1. D and 3D Cell Culture

Rat hepatoma cells, McA Rh7777, were obtained from the American type culture collection (ATCC). The cell was maintained in DMEM (Sigma Aldrich, St. Louis, MI, USA), supplemented with 10% FBS, 1% penicillin streptomycin (PAA, Linz, Austria), and 1% sodium pyruvate (PAA, Linz, Austria), at 37 °C in 5% CO^2^. Sample were harvested by trypsinization, and they were subcultured twice weekly.

The HepG2 + LX-2 spheroids were generated as previously described [14]. Briefly, HepG2 cells and LX-2 (ATCC) cells at a 24:1 ratio were placed into 96-well round-bottomed ultra-low attachment plates (Corning) at 2000 cells/well in MEM supplemented with 10% FBS. They were incubated at 37 °C in a humidified atmosphere of 5% CO_2_ and grown for a total of 96 h.

Briefly, for the generation a liver organoid model composed of primary human hepatocytes (PHH), cryopreserved primary human hepatocytes (donor BGW-M00995-P, BioIVT, Belgium, male, 50 years of age, BMI 20.4) were used [15,16]. Cells were seeded on ultra-low attachment 96-well plates (Corning) at 2000 viable cells per well onto 100 µL of serum containing complete medium. Plates were centrifuged at 100× *g* for 5 min. Once the cells collect at the bottom, self-aggregation causes formation of organoids. At day 1 after seeding, 100 µL of serum-free maintenance medium was added to make a total of 200 µL per well. Next, every 48 h, 50% of the media was replenished for fresh serum-free medium until day 7. The volume of spheroids and organoids were determined using the following formula: 4/3 π r3, where “r” was the mean of the long and short diameter of the spheroid divided by 2.

### 2.2. Treatments

Bergamot PF extract as previously prepared [17] and characterized for polyphenol content was provided by Herbal and Antioxidant Derivatives srl. (Polistena, RC, Italy). As shown in the UV–HPLC chromatogram (Appendix A), the main flavonoids identified in bergamot PF extract were neoeriocitrin (132,517 µg/g), naringin (141,385 µg/g), and neohesperidin (134,119 µg/g), accounting for 40% of 1 gr of powder.

We calculated the bergamot PF extract amount to be use in our experiments based on Parafati et al. [13] by using the equation 50 mg × 70 kg/5000 mL = 0.7 mg/mL, and by assuming a 80–100% bioavailability. In our experiments, we used three order of magnitude less as the highest dose.

For the 2D culture, McA Rh-7777 cells were treated for 24 h with 50 µM of oleic acid (Sigma Aldrich, St. Louis, MI, USA) conjugated to fatty acid-free bovine serum albumin (BSA) and bergamot PF extract dissolved in the cell media at concentrations of 0.001, 0.001, 0.1, and 1 µg/mL.

Instead, HepG2 + LX-2 spheroids, 24 h after seeding, were treated only with bergamot PF extract dissolved in the cell media at a concentration of 1 µg/mL for 72 h. The treatment was refreshed every 48 h.

Human liver organoids, 24 h after seeding, were treated only with bergamot PF extract dissolved in the cell media at a concentration of 1 µg/mL for 6 days. The treatment was refreshed every 48 h.

### 2.3. Cell Viability Assay

For the 2D culture, McA Rh7777 cells were seeded at a density of 1 × 10^4^ cells/well in 96-well plates. Cell viability was determined by 3-(4,5-dimethylthiazol-2-yl)-2,5-diphenyltetrazolium bromide (MTT) assay. Briefly, MTT (Sigma, St. Louis, MO, USA) solution (5 mg/mL) was added to each well and incubated at 37 °C for 4 h. The supernatants were then removed and replaced by 100 mL of DMSO. The optical density (OD) was measured at a wavelength of 570 nm.

For HepG2 + LX-2 spheroids and human liver organoid, total cellular adenosine triphosphate (ATP) was measured in the Cell-Titer-Glo^®^ 3D cell viability assay (Promega, Madison, WI, USA) according to the manufacturer’s instructions. Briefly, single spheroids were transferred into a white 96-well assay plate (Corning, NY, USA) with 50 µL PBS. Then, 50 µL of assay reagent was added to each sample well and vigorously mixed to allow the reagent to penetrate the spheroids, after which the plate was incubated at room temperature for 20 min in darkness. Then, the plate was placed in the SpectraMax i3 (Molecular Devices) counter and luminescence was measured using the SoftMax Pro 6.3 software (San Jose, CA, USA).

### 2.4. Quantification of Intracellular Neutral Lipid Content

For evaluate intracellular lipid content, McA Rh-7777 cells were seeded in a coverslip at a density of 5 × 10^4^ cells/well in 24-well plates. After treatment, the cells were washed with PBS, and fixed with 2% paraformaldehyde for 5 min. Intracellular lipids were stained with Oil Red O solution (Sigma, St. Louis, MO, USA) for 20 min. Cell nuclei were stained with Mayer for 5 min. All the staining procedure was carried out at room temperature by protecting the samples from the direct light. Images were acquired with an Olympus microscope (BX53) and high-resolution Olympus digital image camera (XC50). The images were quantified using ImageJ software (v.1.52h, NIH).

Instead, HepG2 + LX-2 spheroids and human liver organoid after treatment were fixed with 10% *w/v* paraformaldehyde (Sigma-Aldrich) for 2 h, then incubated with 20% *w/v* sucrose in phosphate buffered saline (PBS)(Lonza, Rome, Italy) overnight, washed three times with PBS and embedded with OCT Cryomount (Histolab, Västra Frölunda, Sweden). Then, 8-µm-thick sections were made using a cryostat (Leica, Wetzlar, Germany) and transferred into glass slides and stored at −80 °C for 1 h, after which ORO staining was performed. Nuclei were stained using DAPI. Images were acquired using an Axio KS 400 Imaging System and AxioVision 4.8 software (Zeiss) at 20X. The ORO-stained area was normalized to the number of DAPI-stained nuclei and quantified using ImageJ (v.1.52h, NIH).

### 2.5. Lipid Assay

For evaluate intracellular triglyceride content, McA Rh-7777 cells were seeded in a coverslip at a density of 5 × 10^4^ cells/well in 24-well plates. Cells were then washed with PBS, and fixed with 2% paraformaldehyde for 5 min. Cell nuclei were stained with DAPI for 5 min. Triglyceride content was stained with AdipoRed lipid assay (Lonza, Walkersville, MD, USA) for 10 min at 37 °C. All of the staining procedure was carried out by protecting the samples from the direct light. Images were acquired using an Axio KS 400 Imaging System and AxioVision 4.8 software (Zeiss) at 20X magnification. The images were quantified using ImageJ software (v.1.52h, NIH).

### 2.6. Real Time-PCR

The McA Rh7777 cells were seeded at a density of 400,000 cells/well in 60 mm culture dishes. Total RNA from cells were extracted with TRIzol reagent (Life technologies, Cambridge, UK) according to the manufacturer’s instructions. cDNA was synthesized from 1 µg total RNA, using a high-capacity cDNA Reverse Transcription Kit (Applied Biosystems, Foster City, CA, USA). The mRNA expression of *Srebp-1c, Acox1*, and *Pparα* were quantified by real time-PCR using SYBR^®^ Green dye (SsoAdvanced Univ SYBR Grn Suprmix, Biorad, Milan, Italy) (See Appendix A).

The RNA from McA Rh7777 cells were extracted with the RNeasy Plus Mini Kit (Qiagen, Hilden, Germany) and reverse transcribed using a high-capacity cDNA reverse transcription kit (Thermo Fisher Scientific, Monza, Italy) according to the manufacturer’s instructions. Gene expression was assessed by real-time qPCR using master mix (Life Technologies) and TaqMan probes for *Ucp2, Atg7, ApoB, Mttp*, and *β-Actin* according to the manufacturer’s protocol. All reactions were performed in triplicate. Data were analyzed using the 2^−ΔΔCt^ method normalized to *β-Actin* (See Appendix A).

The RNA from spheroids were extracted with the RNeasy Plus Mini Kit (Qiagen, Hilden, Germany) and reverse transcribed using a high-capacity cDNA reverse transcription kit (Thermo Fisher Scientific) according to the manufacturer’s instructions. Gene expression was assessed by real-time qPCR using master mix (Life Technologies) and TaqMan probes for *SREBP-1C, ACOX1, PPARα, UCP2, ATG7*, and *β-ACTIN* according to the manufacturer’s protocol. All reactions were performed in triplicate. Data were analyzed using the 2−^ΔΔCt^ method normalized to β-Actin (See Appendix A).

### 2.7. Statistical Analysis

Data are represented as mean ± standard deviation (SD) of at least three independent experiments and analyzed using a two-tailed Student’s t-test, linear regression, and nonparametric Mann–Whitney test. The *p*-values less than 0.05 were considered significant. Statistical analysis was performed with GraphPad Prism 5.0.

## 3. Results

### 3.1. Incubation with Bergamot PF Extract Decreases Intracellular Triglycerides in McA Rh7777

To ensure that bergamot PF extract does not affect cell viability, McA Rh7777 cells were incubated with 50 µM of oleic acid and an increasing amount of bergamot PF extract (0.001, 0.01, 0.1, and 1 µg/mL) for 24 h. Given that bergamot PF extract was dissolved directly in the media, cell media was used as a negative control. Cell viability was measured by 3-(4,5-dimethylthiazol-2-yl)-2,5-diphenyltetrazolium bromide (MTT) assay. This assay showed that bergamot PF extract did not affect cell viability at 24 h as compared to the control (Appendix A).

To evaluate the neutral intracellular lipid content in hepatocytes, we incubated McA-Rh7777 cells with increasing concentrations of bergamot PF extract by using Oil Red O staining. Cells were incubated with increasing amount of bergamot PF extract (0.001, 0.01, 0.1, and 1 µg/mL) and after 24 h the intracellular lipid content was examined. Bergamot PF extract resulted in a dose-dependent lowering of intracellular neutral lipids as compared to the control (*p* = 0.001) (Figure 1A,B). At any given dose of bergamot PF extract incubation there was a reduction in the lipid content, specifically *p* = 0.02, *p* = 0.01, and *p* = 0.0008, respectively, except for the 0.1 µg/mL (Figure 1B).

Intracellular neutral lipids are triglycerides and cholesterol esters. To understand what neutral lipid species bergamot PF extract affects we examined cells using AdipoRed assay. Cells were incubated with increasing amount of bergamot PF extract (0.001, 0.01, 0.1, and 1 µg/mL) and, after 24 h, the intracellular triglycerides were measured. Bergamot PF extract incubation resulted in a dose-dependent reduction in the intracellular triglyceride content (*p* = 0.001) (Appendix A). Furthermore, we observed a decrease in the intracellular triglyceride content after incubation with 1 µg/mL bergamot PF extract (*p* < 0.05) (Appendix A).

### 3.2. Bergamot PF Extract Increases Expression Levels of Gene of ß-Oxidation and Lipophagy in McA Rh7777

To understand why incubation of bergamot PF extract reduces intracellular triglycerides, we measured the mRNA levels of gene involved in the ß-oxidation pathway, namely *Acox1*, *Pparα*, and *Ucp2* in McA Rh7777 cells incubated with increasing concentrations of bergamot PF extract (0.001, 0.01, 0.1, and 1 µg/mL). After 24 h of incubation, we observed a dose-dependent increase in *Acox1* (*p* = 0.0001), *Pparα* (*p* = 0.001), and *Ucp2* (*p* = 0.009) (Figure 2A–C).

Intracellular hepatocyte lipid content levels are a function of intracellular triglyceride synthesis, and are utilized via beta-oxidation and in very low-density lipoprotein secretion. Therefore, we examined mRNA levels of a gene involved in triglyceride synthesis, namely Srebp-1c, in McA Rh7777 cells incubated with increasing concentrations of bergamot PF extract (0.001, 0.01, 0.1, and 1 µg/mL). After 24 h incubation, we observed a dose-dependent increase in Srebp-1c (*p* = 0.01) (Figure 2D).

After this, we examined mRNA levels of genes involved in very low-density lipoprotein secretion, namely *ApoB* and *Mttp* in McA-Rh7777 cells incubated with increasing concentrations of bergamot PF extract (0.001, 0.01, 0.1, and 1 µg/mL). After 24 h incubation, we observed that bergamot PF extract does not influence the levels expression of *ApoB* and *Mttp* (Appendix A).

Finally, we examined mRNA levels of genes involved in lipophagy, namely *Atg7* in McA-Rh7777 cells incubated with increasing concentrations of bergamot PF extract (0.001, 0.01, 0.1, and 1 µg/mL). After 24 h incubation, we observed a dose-dependent increase in *Atg7* (*p* = 0.01) (Figure 2E).

### 3.3. Bergamot PF Extract Incubation Reduces Intracellular Neutral Lipid Content and Increases Beta-Oxidation in 3D HEPG2/LX2 Spheroids

To confirm our results in a different cell type and in a more physiological system, we tested incubation with bergamot PF extract in 3D spheroids generated by immortalized hepatocytes (HEPG2) and hepatic stellate cells (LX2) in a physiological ratio of 24:1 (14). Consistently, incubation with bergamot PF extract resulted in a reduction in the intracellular neutral lipid content (*p* = 0.03, Figure 3A–C) without affecting spheroid volume and viability (Figure 3D,E).

Next, we measured mRNA levels of *ACOX1, PPARα, UCP2, SREBP-1C*, and *ATG7* in immortalized spheroids incubated with 1µg/mL bergamot PF extract for 72 h. After 96 h incubation, we observed an increase in *ACOX1* (*p* = 0.03) and *PPARα* (*p* = 0.03) (Figure 4A,B).

Next, we examined mRNA levels of genes involved in triglyceride synthesis, namely *SREBP-1C* in immortalized spheroids incubated with 1µg/mL bergamot PF extract for 72 h. After 96 h incubation, we observed that bergamot PF extract did not influence the levels of expression of *SREBP-1C* (Figure 4D).

### 3.4. Bergamot PF Extract Incubation Reduces Intracellular Neutral Lipid Content in Human Liver Organoid

To further confirm the effect of bergamot PF extract incubation, we generated a human liver organoid by using primary human hepatocytes from a single donor. Briefly, a total of 3000 cryopreserved primary human hepatocytes were seeded on ultra-low attachment 96-well plates. The day after seeding, cells were incubated with 1 µg/mL bergamot PF extract for 6 days. Consistent with the previous experiments, bergamot PF extract incubation resulted in a reduction in the intracellular neutral lipid content (*p* = 0.003, Figure 5A–C) without affecting organoid volume and viability (Figure 5D–E). Results were virtually identical in two further independent experiments (Appendix A).

## 4. Discussion

The main finding of this study is that incubation with bergamot PF extract results in a reduction in intracellular lipid content in human hepatocytes that may potentially be due to an increase in beta-oxidation.

Our bergamot PF extract contained 40% of neoeriocitrin, naringin, and neohesperidin. Administration of these polyphenols resulted in a reduction in liver fat content in humans and rats. To replicate these data, we incubated rat hepatic cells with bergamot PF extract and examined the intracellular lipid content. Incubation with bergamot PF extract resulted in a dose-dependent lowering of intracellular neutral lipids and triglycerides, except for the 0.1 µg/mL treatment (Figure 1B).

Next, we examined the metabolic pathways responsible for the intracellular homeostasis of triglycerides, and we observed an increase in genes involved in beta-oxidation. In the McA-rh7777 cell culture, consistent with the previous study in rats, we observed an increase in *Atg7*, a gene involved in autophagy/lipophagy, a fundamental process involved in human liver disease [18]. However, we also saw a modest increase in Srebp1c, the master regulator of triglyceride synthesis that may be interpreted as an adaptive mechanism to maintain intracellular lipid homeostasis.

To replicate our data in human cells, we used 3D spheroids composed of immortalized human hepatic and hepatic stellate cells. Bergamot PF extract resulted in a reduction in intracellular neutral lipids and an increase in beta-oxidation. However, in this model there was no difference in the synthesis of triglycerides and lipophagy. These data may suggest a species-specific effect of bergamot PF on autophagy in rats as opposed to humans. Species-specific studies are warranted to clarify this observation.

To further confirm the beneficial effect of bergamot PF, we used a liver organoid model composed by human primary hepatocytes. Consistent with previous results, in human and rat hepatoma cells, incubation of bergamot PF extract resulted in a reduction in intracellular neutral lipids. These data show a robust and reproducible effect across hepatocytes derived from three different donors.

Bergamot extract contains compounds with a statin-like effect [9,10,11]. Previous studies show that statin reduces liver fat, inflammation, and fibrosis, most likely by reducing the synthesis of triglycerides [19]. Our data show that bergamot PF, a specific fraction of the bergamot extract, reduces intracellular triglycerides potentially by increasing beta-oxidation. Taking all this together, the administration of bergamot extract may potentially be beneficial in reducing liver fat content via two distinct and synergistic intracellular pathways.

Our study has some limitations. Polyphenols contained in bergamot PF extract are chemically unstable at neutral pH values [20,21]. As a consequence, we cannot exclude that the observed effects are due to bergamot PF extract metabolites. Moreover, approximately 50% of the bergamot PF extract was not composed by polyphenols. Therefore, we the observed effects may derive from other chemical species.

In conclusion, treatment with bergamot PF results in a reduction in hepatocyte intracellular neutral lipids. This effect may potentially be mediated by an increase in beta-oxidation.

## Figures and Tables

**Figure 1 nutrients-14-03434-f001:**
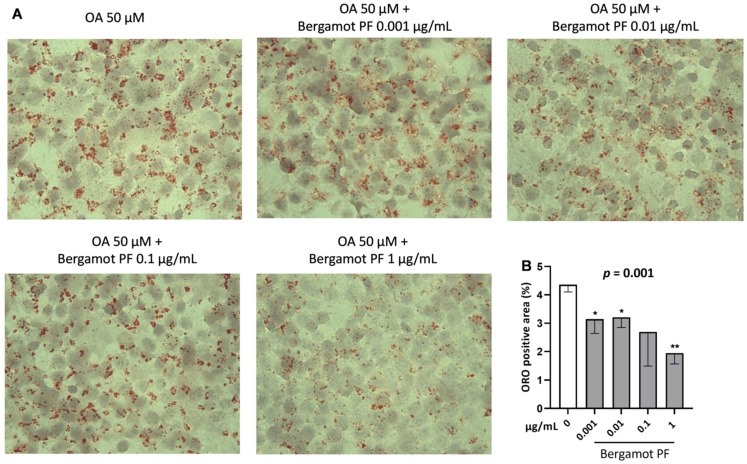
Incubation with bergamot PF extract reduces intracellular lipid content in 2D cultured hepatocytes. Rat hepatoma cell line McA-Rh-7777 was cultured in 2D and incubated with 50 µM of oleic acid and different concentrations of bergamot PF extract (0.001, 0.01, 0.11 µg/mL) in regular medium without FBS for 24 h. Intracellular lipid content was measured by Oil red-O staining (**A**). ORO area quantified by Image J (**B**) and showed a significant decrease in intrahepatic lipid content. Data are shown as mean ± SD of three independent experiments. Statistical analysis methods are as follows: Student`s t-test vs. 0 * *p* < 0.05; ** *p* < 0.001. Linear regression *p* = 0.001. Abbreviations are as follows: PF, polyphenols fraction; FBS, fetal bovine serum; OA, oleic acid; 2D, two dimensional; SD, standard deviation; ORO, Oil red-O.

**Figure 2 nutrients-14-03434-f002:**
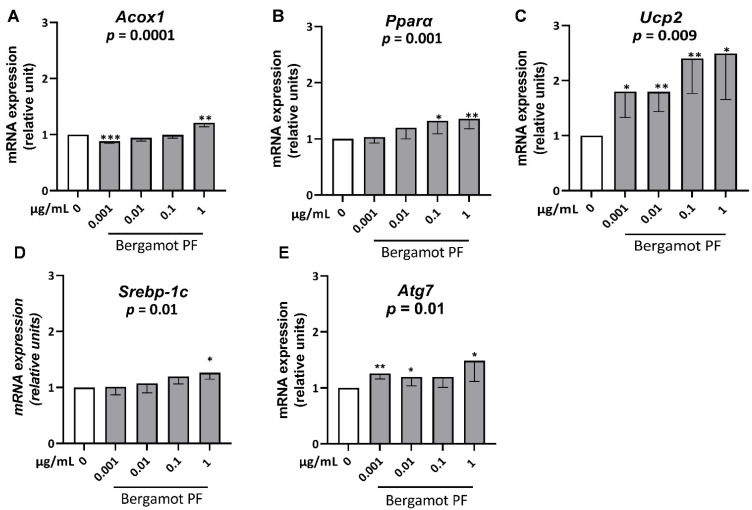
Bergamot PF extract increase regulates *Acox1*, *Pparα*, *Ucp2*, Srebp-1c, and *Atg7* mRNA expression levels in McA Rh-7777 cells. Semi-confluent cultures of rat hepatoma cell line (McA Rh-7777) were incubated with 50 µM of oleic acid and different concentrations of BPF (0.001, 0.01, 0.1, 1 µg/mL) in regular medium without FBS for 24 h. Then, mRNA expression levels of (**A**) *Acox1*, (**B**) *Pparα*, (**C**) *Ucp2*, (**D**) *Srebp-1c*, and (**E**) *Atg7* were measured using real-time PCR. Data were analyzed using the 2^−ΔΔcq^ method and normalized to β-Actin. Data are represented as mean ± SD of three independent experiments. Statistical analysis comprised of Student’s t-test vs. 0 * *p* < 0.05; ** *p* < 0.01; *** *p* < 0.001. Linear trend *p* = 0.001, *p* = 0.001, *p* = 0.009, *p* = 0.01 and *p* = 0.01, respectively. Abbreviations are as follows: PF, polyphenol fraction; *Acox1*, peroxisomal acyl-coenzyme A oxidase 1; *Pparα*, peroxisome proliferator-activated receptor-α; *Ucp2*, mitochondrial uncoupling protein 2; *Srebp-1c*, sterol regulatory element-binding protein; *Atg7*, autophagy related 7; SD, standard deviation.

**Figure 3 nutrients-14-03434-f003:**
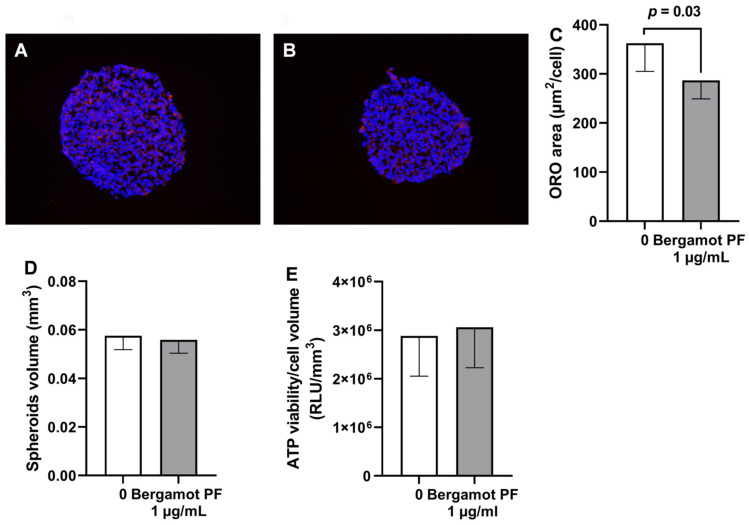
Treatment with 1 µg/mL bergamot PF extract reduces intracellular lipid content in 3D spheroids. (**A**) HepG2 + LX2 cells were cultured as 3D spheroids for a total of 96 h. Initially, 24 h after seeding cells, the media was replaced every 48 h. Objective 20X. (**B**) HepG2 + LX2 cells were cultured as 3D spheroids for a total of 96 h. Initially, 24 h after seeding, the media was supplemented with 1 µg/mL bergamot PF extract and the media was replaced every 48 h. Objective 20X. (**D**) The average volume was calculated by measuring their long and short diameter using the ZEN 2.3 Lite software (Zeiss). (**E**) For both spheroids, cellular ATP levels remained stable between the experimental groups. (**C**) Intracellular lipid content was measured by Oil red-O staining; the results showed a significant reduction after incubation with bergamot PF extract. Data are shown as mean ± SD of three independent experiments. Statistical analysis was as follows: Mann–Whitney non-parametric test vs. 0. *p* = 0.03. Abbreviations are as follows: PF, polyphenols fraction; RLU, relative fluorescence unit; 3D, three dimensional; SD, standard deviation.

**Figure 4 nutrients-14-03434-f004:**
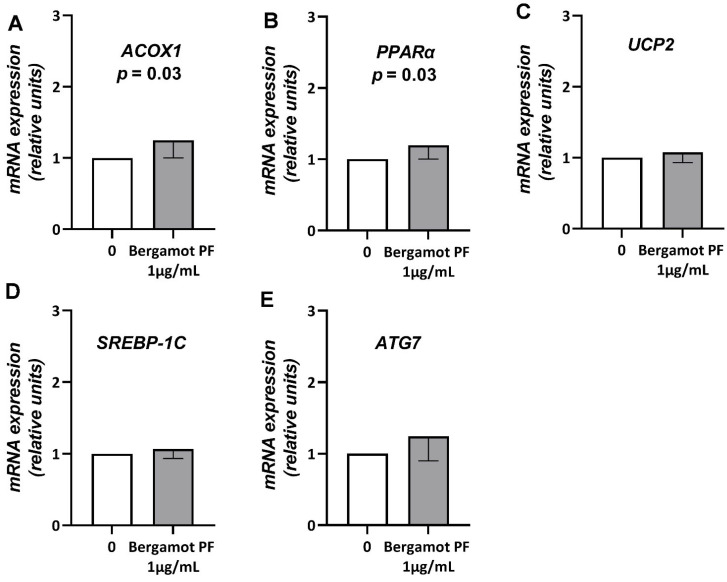
An increase in bergamot PF extract regulates *ACOX1* and *PPARα* mRNA expression levels in 3D spheroids. Here, HepG2 + LX2 cells were cultured as 3D spheroids for a total of 96 h. Then, 24 h after seeding, the media was supplemented with 1 µg/mL bergamot PF extract for 72 h. Then, mRNA expression levels of (**A**) *ACOX1*, (**B**) *PPARα*, (**C**) *UCP2*, (**D**) *SREBP-1c*, and (**E**) *ATG7* were measured using real-time PCR. Data were analyzed using the 2^-^ΔΔcq^ method and normalized to β-*ACTIN*. Data are represented as mean ± SD of three independent experiments. Statistical analysis was as follows: Student’s t-test vs. 0, *p* = 0.03 and *p* = 0.03, respectively. Abbreviations are as follows: PF, polyphenol fraction; *ACOX1*, peroxisomal acyl-coenzyme A oxidase 1; *PPARα*, peroxisome proliferator-activated receptor-α; *UCP2*, mitochondrial uncoupling protein 2; *SREBP-1c*, sterol regulatory element-binding protein; *ATG7*, autophagy related 7; 3D, three dimensional; SD, standard deviation.

**Figure 5 nutrients-14-03434-f005:**
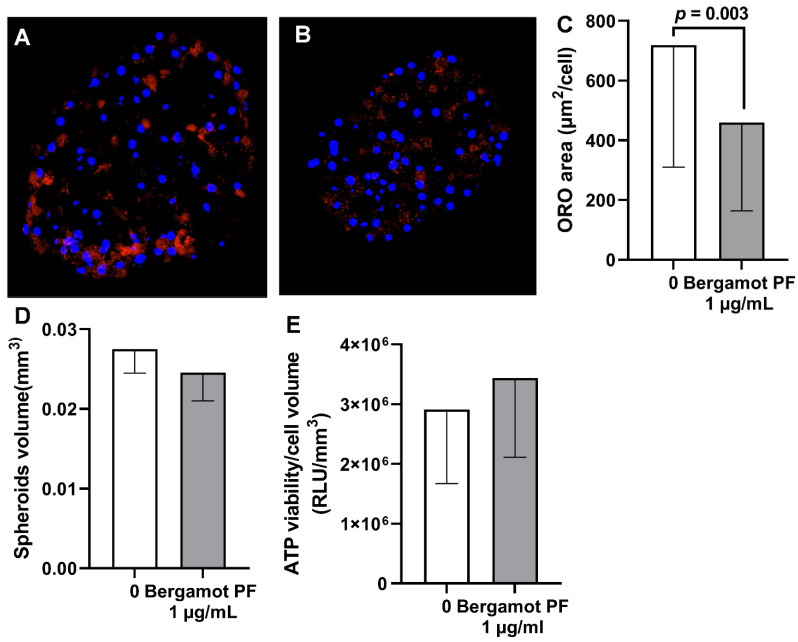
Treatment with 1 µg/mL bergamot PF extract reduces intracellular lipid content in human liver organoid. (**A**) Human liver organoid was cultured as 3D spheroids for a total of 7 days. Initially, 48 h after seeding, the media was replaced every 48 h. Objective 20X. (**B**) Human liver organoid was cultured as 3D spheroids for a total of 7 days. Initially, 48 h after seeding, the media was supplemented with 1 µg/mL bergamot PF extract for an additional 5 days with media replacement every 48 h. Objective 20X. (**D**) The average volume was calculated by measuring their long and short diameter using ZEN 2.3 Lite software (Zeiss). (**E**) For both spheroids, cellular ATP levels remained stable between the experimental groups. (**C**) Intracellular lipid content measured by Oil red-O staining showed a significant reduction after incubation with bergamot PF extract. Data are shown as mean ± SD of three independent experiments. The *p* values were calculated by using a Mann–Whitney non-parametric test. Abbreviations are as follows: PF, polyphenols fraction; 3D, three dimensional; SD, standard deviation.

## Data Availability

Not applicable.

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
