# Peer review of "Bergamot Polyphenol Extract Reduces Hepatocyte Neutral Fat by Increasing Beta-Oxidation"

_nutrients, 2022, doi:10.3390/nu14163434_

Round 1
Reviewer 1 Report
Abstract
Please improve the objective. This study doesn’t elucidate mechanisms.
Introduction
- The introduction should not include the manuscript conclusion (lines 54-56).
Please improve the objective. This study doesn’t elucidate mechanisms.
Materials and methods
- Line 58: 2.1.2 D and 3D cell à 2.1 2D and 3D cell.
Results
- In figure 1, the concentration 0.1 ug/ml was not significant in reducing lipid content? Please discuss.
- Line 193-194: “Bergamot PF incubation resulted in a dose-dependent reduction in the intracellular triglyceride content (P=0.001) (Figure S2 A-B).” This is not significant; only the 1 ug/ml dose was effective.
- Line 209-210: there is a typing error; please review.
Discussion and conclusion
- Discussion needs to be deepened
- Lines 293-295: This sentence is very categorical since association does not mean causality.
- Lines 305-307: this result was not significant.
- Lines 312-313: Are the doses equivalent?
- Please discuss the concentrations of bergamot polyphenols used. Are the amounts equivalents to doses used in in vivo studies?
- Authors conclude that bergamot polyphenols reduce intracellular lipids by inducing specific molecular pathways (B-oxidation). However, they don't demonstrate causality, only association, since they don't block nor exacerbate particular pathways. I suggest improving the conclusion according to the paper's methods and results.
Reviewer 2 Report
The work by Mirarchi and colleagues entitled “Bergamot polyphenol extract reduces hepatocyte neutral fat by increasing beta-oxidation” describes the effect of bergamot (poly)phenol extract in 2D and 3D cultures (McA Rh-777 cell model, HepG2+LX-2 spheroids and liver organoids) on neutral lipid content and gene expression levels which were complemented by cell viability assays.
The literature is prolific on the health benefits of (poly)phenol intake and hence additional studies able to support and introduce additional mechanisms by which dietary (poly)phenols contribute to improved understanding are highly desirable. The effect of (poly)phenol intake on the lipid metabolism pathways is for sure an interesting topic. While the authors increased the complexity of their experiments, they have neglected some basic aspects of (poly)phenol research (please see below) which hinders validation of the findings reported and conclusions drawn and hence the manuscript is not acceptable in its present form.
Major concerns:
11) The authors have implemented a 24hr incubation period of bergamot extract for the McA Rh-777 cell model, a 72h treatment for the spheroids and 6 days treatment for organoids (see Section 2.2). It is not clear in the manuscript the rationale behind this. If different incubation treatments were carried out in different cultures then the different results (trends) are expected.
22) The authors have disregarded the acknowledged poor chemical stability of plant (poly)phenols at neutral pH values such as those found in cell cultures (please see works by Zhou, Q., Chiang, H., Portocarrero, C., Zhu, Y., Hill, S., Heppert, K., … Kissinger, P. (2003). American Chemical Society (ACS)., 20(3), 83-86; Li, N., Taylor, L. S., Ferruzzi, M. G., & Mauer, L. J. (2012). J Agric Food Chem, 60, 12531; Sang, S., Lee, M. J., Hou, Z., Ho, C. T., & Yang, C. S. (2005). J. Agric. Food Chem., 53(24), 9478–9484; Xiao, J., & Högger, P. (2015). J. Agric. Food Chem., 63(5), 1547–1557; Zeng, L., Ma, M., Li, C., & Luo, L. (2017) International Journal of Food Properties, 20(1), 1–18.). To relate the molecular traits of bergamot (poly)phenols to the observed results in cultures the authors need to introduce a HPLC step for the characterization of bergamot polyphenols in the culture media at the end of the incubation period. It is unlikely that flavonoids (neoeriocitrin, naringin and neohespiridin) survive intact for such long incubation periods in neutral medium (even if refreshed every 48hr). The authors need to show that the observed effects are due to BPF and not to BPF degradation products. Without this step the authors can hardly claim that the bergamot polyphenol treatment markedly reduces the levels of intracellular neutral lipids in 2D and 3D cultures.
33) Similarly, by studying the effects of bergamot PF (BPF) extract in cell cultures, the authors have not taken into account the biotransformation (metabolization) of bergamot (poly)phenols upon ingestion. In reality, the compounds present in the extract will not be the same as those found in vivo and so it would have been much more physiologically relevant to study the effect of digestion on bergamot extract PF and use this information to improve the 2D and 3D culture designs. Unless corroborated by the available literature, it is unclear how neoeriocitrin, naringin and neohesperidin reach the liver in its “unmodified” form to exert the health benefits as these will reach the liver after gut metabolization.
Other concerns:
44) The use of bergamot extract is a major weakness. The bergamot (poly)phenol extract has been previously characterised (containing flavonoids neoeriocitrin (370 ug/mL), naringin (520 ug/mL) and neohesperidin (310 ug/mL) as stated in lines 81-83 of the manuscript, according to ref 17 (Gliozzi and colleagues, 2013). However, the work by Gliozzi and colleagues (2013) is omissive in important details such as the amount of bergamot fruits used in squeezing step to produce the BPF extract. Also, it does not state information on the amount of BPF extract used in the flavonoid characterisation. Assuming that flavonoid characterisation was done in the 500mg BPF tables (Gliozzi et al., (2013) dissolved in 1mL solution, it means that flavonoids totalise 1,2mg leaving to wonder what the remaining material (498.8mg) present in the BPF composition is. It is therefore difficult to infer whether the observed effects are due to bergamot main flavonoids or other compounds present and unaccounted for.
55) In the manuscript the authors make no mention as to what extract this relates to. This important piece of information is detailed in ref 17 requiring the reader to chase other papers. A simple mention to aqueous extract is helpful.
66) Line 80 where it states “Bergamot PF…” should state “Bergamot PF extract…” as the extract was used in cultures and not the individual flavonoids isolated from bergamot extract.
77) For the various assays, the number of replicates is not mentioned neither in the experimental section or the figure captions.
Round 2
Reviewer 2 Report
The authors have considered major points raised by this reviewer and ammemded the issues appropriately in the text.